# Chemotherapy Side-Effects: Not All DNA Damage Is Equal

**DOI:** 10.3390/cancers14030627

**Published:** 2022-01-26

**Authors:** Winnie M. C. van den Boogaard, Daphne S. J. Komninos, Wilbert P. Vermeij

**Affiliations:** 1Princess Máxima Center for Pediatric Oncology, Heidelberglaan 25, 3584 CS Utrecht, The Netherlands; w.vandenboogaard@prinsesmaximacentrum.nl (W.M.C.v.d.B.); d.s.j.komninos@prinsesmaximacentrum.nl (D.S.J.K.); 2Oncode Institute, Jaarbeursplein 6, 3521 AL Utrecht, The Netherlands

**Keywords:** chemotherapy, DNA damage, cancer treatment, premature aging, cancer survivors

## Abstract

**Simple Summary:**

The number of children and adults with cancer that are completely cured is still increasing thanks to effective anti-cancer therapy, but they may be confronted with the negative lasting effects of the treatment later in life. In this review, we provide an overview of major clinical symptoms and toxic side-effects observed in cancer survivors. We describe which types of anti-cancer treatments—primarily DNA-damaging chemotherapeutics—might cause these toxicities and what the (potential) underlying mechanisms are. These treatments not only damage cancer cells in their attempt at tumor killing but also harm healthy cells and tissues. Observations of side-effects in cancer patients and survivors strengthen the hypothesis that the primary induced DNA damage can lead to varying toxicities while also accelerating features of aging, depending on type and dose of chemotherapeutic, clearing method, and affected organ.

**Abstract:**

Recent advances have increased survival rates of children and adults suffering from cancer thanks to effective anti-cancer therapy, such as chemotherapy. However, during treatment and later in life they are frequently confronted with the severe negative side-effects of their life-saving treatment. The occurrence of numerous features of accelerated aging, seriously affecting quality of life, has now become one of the most pressing problems associated with (pediatric) cancer treatment. Chemotherapies frequently target and damage the DNA, causing mutations or genome instability, a major hallmark of both cancer and aging. However, there are numerous types of chemotherapeutic drugs that are genotoxic and interfere with DNA metabolism in different ways, each with their own biodistribution, kinetics, and biological fate. Depending on the type of DNA lesion produced (e.g., interference with DNA replication or RNA transcription), the organ or cell type inflicted (e.g., cell cycle or differentiation status, metabolic state, activity of clearance and detoxification mechanisms, the cellular condition or micro-environment), and the degree of exposure, outcomes of cancer treatment can largely differ. These considerations provide a conceptual framework in which different classes of chemotherapeutics contribute to the development of toxicities and accelerated aging of different organ systems. Here, we summarize frequently observed side-effects in (pediatric) ex-cancer patients and discuss which types of DNA damage might be responsible.

## 1. Introduction

In the current era of medical treatments, the cure rate of cancer has reached approximately 80% in children [1,2,3] and 67% in adults [4]. Although many treatment modalities are being developed, the main components of therapy still include surgery, radiotherapy, and chemotherapy. During the 19th and the first part of the 20th century, surgical resection was the only possible treatment for cancers. Radiotherapy became available around 1900 but proved, like surgery, unsuccessful in many patients. Cure rates improved after the invention of chemotherapy. Chemotherapy was first introduced shortly after World War II with nitrogen mustard as a treatment modality against lymphomas, leukemias, and Hodgkin’s disease [5]. Shortly after, aminopterin, a predecessor of methotrexate, was prescribed to children with acute leukemia [6]. More chemotherapeutic drugs were developed during the 20th century, which are nowadays used in different combinations in various treatment schedules depending on a.o. tumor type, risk stratification group, and genetic landscape [7,8]. Treatment protocols are being updated continuously to increase survival rates. In addition, the focus of treatment optimization has shifted to diminish therapy-related toxicity. Some of the most common side-effects induced by anti-cancer treatments are secondary (therapeutic-induced) malignancies, nephro-, hepato-, neuro-, cardio-, and ototoxicity [9,10,11,12,13,14]. To kill tumor cells, chemo- and radiotherapy are frequently used that mainly target DNA, but these therapies also elevate DNA damage in (surrounding) healthy tissue, causing toxicities and accelerated aging [10,12,14,15].

Reduction of therapy-induced side-effects would be of benefit to all age groups. Since life expectancy has increased over the last decades, more cancer survivors and especially children still have many years to live with a lowered quality of life after being cured from cancer. Generally, children tolerate higher drug doses (in mg per m^2^ body surface area) than adults, likely due to higher drug clearance rates and larger volumes of drug distribution [16]. These differences are perhaps caused by better functioning of detoxifying organs, liver, and kidneys, and larger total body water percentage compared to adults [16]. As drug dose is guided by the maximum tolerated dose, children are often treated with higher dosages of the same drug compared to adults [16]. This might contribute to varying degrees of toxicity in pediatric patients compared to adult patients [17]. As most pediatric cancer patients are subjected to combination therapy, it is hard to link the observed side-effects to one specific chemotherapeutic. Knowledge of medications used during cancer treatment, their mechanism of action, and toxicities occurring in different organs is required to timely observe development of toxicity, adjust treatment accordingly, and prevent further deterioration. The conventional chemotherapeutics can be divided into agents that do or do not lead to DNA damage. In case of DNA damage, DNA repair mechanisms can be activated according to the type of damage inflicted [18] (Figure 1). However, the doses of cytostatic agents are usually high, making it impossible for the cells to repair the damage adequately. In case of non-DNA-damaging agents, cellular processes involved in duplication or proliferation are inhibited, preventing further expansion, ultimately leading to cell death [19]. This review focuses on the classical chemotherapeutics and their consequential side-effects observed in patients (Table 1). Organ-specific toxicities are described, alongside their causative agents and (potential) underlying mechanisms.

## 2. Chemotherapy Side-Effects Observed in Cancer Survivors

### 2.1. Nephrotoxicity

Many chemotherapeutic drugs are cleared from our body by the kidneys [27,31,69]. The two major drug excretion routes are via glomerular filtration and tubular secretion [24]. Renal tissue is therefore exposed to higher concentrations of most drugs than blood and likely other organs, frequently leading to acute kidney injury (AKI) or late-life nephrotoxicity [24,27,31,42,69].

#### 2.1.1. Alkylating Agents

Frequently used chemotherapy subtypes are alkylating agents, such as platinum agents, cyclophosphamide, and ifosfamide. Platinum agents, used for multiple cancers, mostly injure proximal tubule cells, as uptake of these chemotherapeutics by proximal and distal tubular cells is necessary for active secretion into urine, resulting in accumulation of the drug in these cells [27,70,71]. The most common platinum drug, cisplatin, is also highly toxic as it easily reacts with DNA [33]. Cisplatin is taken up via membrane transporters such as organic cation transporter 2 (OCT2) and high affinity copper uptake protein 1 (CTR1), which are highly expressed on proximal tubular cells [72,73]. As soon as cisplatin is transported into the cell’s cytoplasm, it becomes reactive by the replacement of one or two of its chloride groups. After this process, also known as aquation, cisplatin will bind covalently with the DNA, forming intrastrand or interstrand crosslinks [15,74] (Figure 1). Crosslinks between two strands of DNA are more harmful, since these block DNA replication and are less efficiently tackled by the DNA repair machinery [15]. Due to the absence of an intact template strand, crosslink repair is prone to errors, which can lead to double-strand breaks, cell-cycle arrest, and eventually senescence or programmed cell death, contributing to aging and development of secondary cancers [74,75]. Other platinum agents, such as carboplatin or oxaliplatin, have one of the two chloride groups replaced by carboxylate or cyclobutane, respectively, making these compounds less reactive [24]. Concentrations of cisplatin are approximately 5-fold higher in proximal tubular cells than in blood [42,76], and as both the renal tubules and glomeruli are actively involved in cisplatin elimination, complications may arise in both structures [42]. Besides directly causing DNA damage, platinum agents can also increase reactive oxygen species (ROS), generating cellular damage, ER stress, and stimulating the mitochondrial apoptotic pathway [25,69,77]. On a microscopic level, exposure to high doses of cisplatin leads to inadequate reabsorption of small molecules (e.g., glucose and amino acids) also known as Fanconi syndrome (FS), interstitial oedema, and dilation of the proximal and distal tubules [27,31]. Potential clinical manifestations of cisplatin-induced kidney damage are reduced creatinine clearance, impaired magnesium reabsorption, decreased glomerular filtration rate (GFR), AKI, distal renal tubular acidosis, thrombotic microangiopathy (TMA), and chronic renal failure [21,31,78].

Other alkylating agents, such as ifosfamide, used against sarcomas, lymphomas, and testicular/ovarian cancers, appear to be nephrotoxic mostly at high doses [31]. Toxic side-effects can be caused by the agent itself or by its metabolite chloroacetaldehyde. Like platinum agents, ifosfamide is taken up by the proximal tubule cells, causing tubular dysfunction and FS [21,27,29]. Due to acute tubular necrosis (ATN) and impaired sensitivity to anti-diuretic hormone (ADH), the kidneys fail to concentrate urine. This so-called nephrogenic diabetes insipidus (NDI) will result in polyuria causing severe dehydration and electrolyte imbalances [29,79].

#### 2.1.2. Antimetabolites

Antimetabolites (e.g., methotrexate, pemetrexed, and gemcitabine) are competitive inhibitors of metabolic processes and are predominantly eliminated by the kidneys, leading to nephrotoxicity. Generally, incorporation of antimetabolites in DNA or RNA (precursors) leads to inhibition of DNA and RNA synthesis, halting cell division and triggering cell death [27]. Methotrexate, an antifolic antimetabolite, is one of the most widely used antineoplastic agents against leukemias, lymphomas, and osteosarcomas [27]. This drug and its metabolites precipitate into renal tubule cells, due to insufficient dissolution in urine at low pH, causing crystal nephropathy [27,29,31,33]. This crystal deposition causes intratubular obstruction, leading to renal dysfunction and eventually AKI [33,80]. Methotrexate also transiently decreases GFR due to vasoconstriction of afferent renal arteries, ultimately leading to glomerular dysfunction [33]. Pemetrexed, often used to treat non-small cell lung cancer (NSCLC), is an antimetabolite that inhibits enzymes involved in purine/pyrimidine synthesis [24,27]. This drug is often associated with impaired creatinine clearance, tubular damage, and AKI [24]. Gemcitabine is primarily used against carcinomas in lung, pancreas, bladder, and breast, and its antiproliferative activity also depends on inhibition of DNA synthesis. This occurs as soon as gemcitabine is incorporated in DNA, which leads to termination of DNA elongation by DNA polymerase [81]. Treatment with gemcitabine can cause drug-induced TMA, AKI, and sometimes renal failure, indicating the severe toxic side-effects of this drug [21,24,35].

#### 2.1.3. Anti-Cancer Antibiotics

Anthracyclines, a subgroup of anti-cancer antibiotics also known as topoisomerase II inhibitors, are mostly toxic to the heart (see below); but in addition, the anthracycline doxorubicin also causes nephrotoxicity. This drug is mostly used against breast cancer, aggressive lymphomas, childhood solid tumors, and soft tissue sarcomas and its mechanism of action is predominantly based on DNA intercalation [82]. As a result, DNA and RNA synthesis are blocked and DNA lesions such as double- and single-strand breaks emerge [83,84], causing nephrotic syndrome, glomerular sclerosis, TMA, and AKI [33,36]. Anti-cancer antibiotics often have TMA manifested as hemolytic uremic syndrome (HUS), leading to AKI. Mitomycin is an alkylating agent that inhibits DNA synthesis by interstrand crosslink formation. Its uptake frequently leads to direct damage to the renal parenchyma leading to TMA, HUS, and AKI [33,35].

### 2.2. Hepatotoxicity

The liver’s role in detoxification, drug metabolism, and excretion of waste products partially explains why many cancer therapies are harmful to the liver [38]. However, the specific mechanisms underlying hepatotoxicity of many agents remain to be elucidated. Various therapies are associated with liver toxicity, including irinotecan, cisplatin, oxaliplatin, and irradiation, which can manifest in several ways [40,85]. Hepatitis, cholestasis, and steatosis are some of the most frequent toxic side-effects of chemotherapy [38].

#### 2.2.1. Alkylating Agents

Generally, alkylating agents are less toxic to liver compared to kidneys, likely due to low expression of transporters required for platinum uptake [39,70,72]. One of the most hepatotoxic alkylating agents is cyclophosphamide. It is activated and degraded by the liver via P450 microsomal oxidative mechanisms, generating the cytotoxic acrolein and phosphoramide mustard as by-products [43]. Combined with irradiation, cyclophosphamide toxicity is highly increased [38]. Platinum-based chemotherapeutics and nitrosoureas generate toxic effects in the liver due to uptake by the hepatocytes for metabolism and detoxification [28,43]. This causes accumulation of intra- and interstrand crosslinks and hepatocyte polyploidization, reducing their normal functioning [42]. Occasionally, cisplatin and carboplatin generate cholestasis, steatohepatitis, and veno-occlusive disease (VOD) [38,39]. Oxaliplatin is often used in combination therapy, causing steatosis, VOD, and sinusoidal obstruction syndrome (SOS) [38,39,40]. Nitrosoureas exert their cytotoxicity via both alkylation and carbamoylation of DNA, leading to cell-cycle arrest and cell death [28,43]. Besides these DNA adducts, nitrosoureas also exert their cytotoxic effects via depletion of hepatic stores of glutathione, a strong antioxidant, thereby increasing oxidative injury [39,43,44].

#### 2.2.2. Antimetabolites

Many antimetabolites, such as cytarabine, 5-fluorouracil, methotrexate, and anthracyclines, are metabolized by the liver. Consequently, hepatocytes are exposed to the toxic side-effects of these drugs or their by-products [39]. Cytarabine, or cytosine arabinoside (ara-C), is often used against leukemia and non-Hodgkin lymphomas [39]. It is a pyrimidine nucleoside analogue mimicking cytidine and deoxycytidine, which are necessary for DNA synthesis [43]. Treatment with cytarabine can lead to cholestasis, biliary stricture, and fibrosis [38,39]. 5-fluorouracil (5-FU) is metabolized primarily in the liver and seems to accumulate in hepatocytes, where uracil analogues can interfere with RNA synthesis directly and DNA synthesis by inhibiting conversion of dUMP to dTMP [37,86]. Subsequent dNTP imbalance ultimately leads to DNA strand breaks and apoptosis [87]. However, most toxicity reports were made after intra-venous injections of 5-FU, not with oral administration [38,39,44]. 6-Mercatopurine (6-MP), often used in acute lymphoid leukemia, inhibits purine synthesis and can be incorporated into DNA leading to apoptosis [44]. Most common side-effects of this drug are cholestatic and hepatocellular liver disease [38,43,44]. Methotrexate therapy often leads to liver cirrhosis and fibrosis, and capecitabine, gemcitabine, and floxuridine frequently cause hepatitis and cholestasis [38,39,43].

#### 2.2.3. Topoisomerase Inhibitors

Topoisomerase inhibitors damage DNA by blocking and disturbing the function of topoisomerase in the process of DNA replication. Topoisomerase I binds to DNA and allows it to uncoil by cutting one strand, passing the other one through the single-strand break, and then resealing the cleaved strand. Formation of transient DNA double-strand breaks is performed by topoisomerase II to release tension on the DNA helix [88,89]. Topoisomerase inhibitors block DNA replication by inhibiting re-ligation of the cleaved strand(s), leaving the generated DNA breaks unrepaired [90]. Additionally, increased tension on DNA prevents proper functioning of DNA and RNA polymerases, followed by DNA damage and transcriptional arrest and eventually cell death [91]. Irinotecan, a topoisomerase I inhibitor, is often used for colorectal, lung, and ovarian cancers [90]. It is a prodrug that is converted into the active metabolite SN-38 via carboxylesterases CES1 and CES2. SN-38 is mostly metabolized into an inactive form (SN-38G) by uridine glucuronosyltransferase 1A1 (UGT1A1), which is primarily found in the liver [37]. The exact mechanism of irinotecan-induced hepatotoxicity remains to be elucidated, but it seems that both oxidative stress and mitochondrial dysfunction play a large role. Mitochondria dysfunction likely arises by inhibition of replication of mitochondrial DNA by irinotecan. This can, in turn, lead to increased ROS production by a defective respiratory chain and increased lipid peroxidation [37]. Clinical manifestations of irinotecan hepatotoxicity are steatosis and hepatitis. Etoposide is a topoisomerase II inhibitor, which is mostly excreted by bile and relatively mild for the liver [39,44]. There are, however, a few observations of hepatocellular injury, hepatitis, and cholestasis [39,43].

#### 2.2.4. Other Chemotherapeutics

Anti-cancer antibiotics can be hepatotoxic, especially dactinomycin and mitomycin are identified as hepatotoxic chemotherapeutics, although they differ in mechanism of action [39]. Dactinomycin intercalates with the DNA, blocking RNA and protein synthesis [39]. Mitomycin works as an alkylating and crosslinking agent, mostly inhibiting DNA synthesis, and often leading to cell-cycle arrest and/or apoptosis [43,44]. Both compounds often cause hepatitis and VOD [39]. Another drug class, mitotic inhibitors, such as taxanes and vinca alkaloids, target microtubules. Vinca alkaloids specifically bind tubulin, thereby primarily inhibiting microtubule formation, disrupting mitotic spindle formation, and finally leading to metaphase arrest [92]. Vincristine and vinblastine can cause hepatitis and VOD [39,43].

#### 2.2.5. Radiotherapy

Radiotherapy mostly injures organs in the abdomen [93,94]. Radiation-induced liver disease (RILD) typically occurs within four months after radiotherapy [45]. Irradiation injures the sinusoidal endothelial cells and central vein endothelium, initiating fibrin and collagen proliferation and cloth formation. This results in trapped erythrocytes and vascular congestion, also known as sinusoidal obstructive syndrome (SOS) or VOD [43,44,45]. Hepatic stellate cells, which are activated after endothelial damage, are involved in this process. Namely, besides their engagement in regeneration of hepatocytes, they also secrete pro-inflammatory cytokines and specifically TGF-β is suspected to be involved in subendothelial and hepatic fibrosis in RILD [45]. In addition, radiation damage can lead to increased levels of DNA breaks in hepatocytes driving steatosis [95].

### 2.3. Neurotoxicity

Neurotoxicity is mainly presented with sensory and occasionally sensorimotor characteristics, such as numbness, abnormal sensation, or paresthesia in distal extremities, reduced reflexes, and leg weakness [50,91,96]. This means that the peripheral nervous system (PNS) is most frequently affected, especially sensory neuronal cell bodies in dorsal root ganglia (DRG) of primary neurons [47,48,50], while the central nervous system (CNS) is mostly protected by the blood–brain barrier. Long-term neurotoxicity strongly decreases quality of life. There are, however, no methods to decrease this toxicity, besides lowering treatment dose or instantly terminating treatment upon onset of neurotoxicity [50].

#### 2.3.1. Alkylating Agents

While many alkylating agents can result in some form of neurotoxicity, platinum agents are described as most damaging to the DRG [47,50]. CTR1 is highly expressed on neuronal cells in the DRG, leading to efficient uptake of platinum compounds, likely explaining the toxicity [73]. Cisplatin-induced neuropathy is dose-dependent and usually follows a cumulative dose of 400 mg/m^2^ or higher [48]. Oxaliplatin can, in addition to peripheral sensory neuropathy, cause acute neurotoxicity. In contrast to chronic neuropathy, oxaliplatin-induced acute neurotoxicity does not require dose modification or treatment discontinuation. However, it might contribute to the onset and severity of chronic toxicity [47,50].

#### 2.3.2. Antimetabolites

Treatment with antimetabolites can lead to both PNS and CNS neurotoxicity. Methotrexate, a compound that is able to cross the blood–brain barrier and one of the most neurotoxic chemotherapeutics, can cause acute, subacute, and delayed neurotoxicity [48,49]. Acute encephalopathy is reversible, subacute stroke-like syndrome is typically characterized by focal neurologic deficits, aphasia, and seizures, and the most common chronic toxic side-effect caused by methotrexate is leukoencephalopathy [48,49]. On MRI scans, leukoencephalopathy is recognized by diffuse white matter hyperintensities and parenchymal volume loss. It often leads to cognitive impairment, dementia, ataxia, seizures, and focal deficits [49]. The most concerning toxicity caused by cytarabine is cerebellar dysfunction, or pancerebellar syndrome, which causes nystagmus, dysarthria, and ataxia. This is usually reversible, but in some patients these side-effects become permanent due to cerebellar Purkinje neuron loss [48,49].

#### 2.3.3. Mitotic Inhibitors

Vinca alkaloids, a subtype of mitotic inhibitors, are used in treatment of both non-solid and solid tumors, such as hematologic and lymphatic malignancies, and breast, ovarian, testicular, brain, and NSCLC [47]. Vincristine is the most neurotoxic of all vinca alkaloids, probably due to its high tubulin binding affinity and long half-life [50]. Vinca alkaloid treatment generates detrimental modifications to axonal cytoskeletons by disorientation of microtubules and neurofilament accumulation, which leads to alteration in length, arrangement, and orientation [47,50]. Both these events, as well as direct axonal toxicity, lead to loss of function, which might explain the strong side-effects of vincristine therapy [49,50]. Besides axonal damage, vinca alkaloids can also reduce myelin thickness, shortening of inter-nodal length, and segmental demyelination [47].

Taxanes are another subtype of mitotic inhibitors, and instead of scavenging tubulin resulting in blocked microtubule polymerization, taxanes bind microtubules. This causes aggregate formation, eventually leading to similar complications in large axons, as described with vinca alkaloids [46,50]. Both paclitaxel and docetaxel can cause severe neurotoxicities, such as distal sensory neuropathy, paresthesia, arthralgia-myalgia syndrome (AMS), myopathy, and motor impairment [46,50].

#### 2.3.4. Proteasome Inhibitors

Proteasome inhibitors are a relatively new type of chemotherapeutic drug that decrease protein turnover, leading to protein accumulation, ultimately causing apoptosis. They are mostly used for treatment of multiple myeloma [97]. Bortezomib was the first FDA-approved proteasome inhibitor and is also most neurotoxic [47,50,97]. It is especially damaging to the DRG, where it interferes with transcription, nuclear processing and transport, and cytoplasmic translation of mRNAs. There is evidence of reduced activation of nuclear factor (NF)-κB due to bortezomib-mediated stabilization of the inhibitor IκB and reduced cyclin turnover, affecting cyclin-dependent kinase activity, leading to cell-cycle arrest [47,97]. Bortezomib toxicity is characterized by very painful sensory neuropathy, accompanied by distal sensory loss and hypo- or areflexia [47,50].

### 2.4. Cardiotoxicity

The heart, being the motor of the body, is one of the most essential organs for normal functioning. Decreased functioning of the heart can lead to severely reduced quality of life and in critical cases even death. Dilated cardiomyopathy, rhythm disturbances, and myocardial infarctions are some of the cardiotoxicities that can develop after chemotherapy treatment [98]. Cardiovascular disease is in the top three causes of long-term morbidity and mortality in cancer survivors [99,100], affirming the need for prevention and monitoring of cardiac changes during cancer treatment.

#### 2.4.1. Anthracyclines/Cytostatic Antibiotics

Anthracyclines (e.g., doxorubicin, daunorubicin, epirubicin, and idarubicin) are used in the treatment of hematological and solid tumors and are frequently reported to be cardiotoxic [101]. Development of toxicity is thought to occur from the start of therapy onwards, building up over time in a dose-dependent manner. The main side-effect is (congestive) heart failure, which can be observed shortly after treatment, but also years after therapy is finalized [102]. Structurally each anthracycline drug consists of an anthraquinone group inside a tetracycline ring and a carbohydrate moiety attached to this ring via a glycosidic bond. Multiple mechanisms have been described for anthracycline-induced cardiotoxicity.

Firstly, the quinone group present on anthracyclines is involved in electron transfer reactions, whereby cycling between the quinone and semiquinone form leads to excessive formation of ROS [103,104,105]. Additionally, reaction of the quinone group with O_2_ or H_2_O_2_ with or without the presence of iron can lead to free radical formation [103,106].

Secondly, toxicity of anthracyclines is thought to be higher in cardiomyocytes due to its lower antioxidant capacity compared to other tissues. Lack of natural expression of catalase [107] and superoxide dismutase [108] and anthracycline-induced suppression of glutathione peroxidase [108] result in accumulation of ROS in heart tissue specifically. Subsequent formation of single- and double-strand breaks leads to activation of DNA damage repair, or, in case of excessive damage, apoptosis or senescence.

Anthracycline-induced cardiotoxicity can also occur via anthracycline targeting of topoisomerase II [109]. Topoisomerase II has two isoforms, the α (Top2α) and β (Top2β) variants, which are expressed in proliferating and quiescent cells, respectively. Only Top2β is expressed in heart tissue and its presence is hypothesized to be of importance in anthracycline-induced cardiotoxicity [110].

Thirdly, anthracyclines seem to target mitochondria, of which high levels are present in heart tissue, both structurally and functionally. Anthracyclines can change mitochondrial stability by binding to cardiolipin, a phospholipid present on the inner membrane of mitochondria [111], and reduce transcription of peroxisome proliferator-activated receptor-gamma coactivator 1-α and -β (PGC-1α and PGC-1β) [112], which are required for proper functioning of the electron transport chain.

A more recently discovered effect of anthracyclines is histone eviction [113,114]. The exact mechanism is not yet identified, but subsequent effects include attenuation of the DNA damage response, epigenetic changes, and induction of apoptosis.

#### 2.4.2. Other Chemotherapeutic Agents

Cardiotoxicity is observed after treatment with alkylating agents, but for most of these (cyclophosphamide, ifosfamide, cisplatin) the mechanism behind the specific sensitivity of cardiomyocytes is unknown [51]. For mitomycin, which harbors a quinone moiety, it is thought it can undergo redox reactions, thereby creating ROS contributing to the toxic effects on the heart [115]. The antimetabolite 5-fluorouracil and antimicrotubular agents, paclitaxel and docetaxel, are associated with cardiotoxicity, but the underlying causal processes are not known. For paclitaxel, it is thought the effects on the heart might be induced by its solvent (Cremophor EL), which can lead to induction of IL-8 and activation of MAPK p38 signaling, and not paclitaxel itself [116,117,118].

### 2.5. Hematological Toxicities

Blood parameters, such as number of blood cells, are monitored before and after treatment cycles to reveal occurrence of cytopenia. The main reason to follow changes in blood cell populations is the risk of infection [119,120]. Certain chemotherapeutic agents are known to lead to anemia, neutropenia, and/or thrombocytopenia. When the loss of blood cells has reached a certain threshold, treatment dose is lowered to reduce infection risk [119].

Information on specific chemotherapies causing hematological toxicity is very limited. One study used public data on antineoplastic drugs that were reported to cause neutropenia obtained from the Food and Drug Administration Adverse Event Reporting System [121]. Associations were found between alkylating, platinum, and antimetabolic agents, antineoplastic antibiotics, plant-derived alkaloids, and the occurrence of neutropenia. The increased risk of severe or febrile neutropenia after anthracycline- or platinum-based regimens was replicated in another study specifically performed in older patients [122]. Both regimens were also associated with a relative dose intensity <85%, meaning patients received lower doses than prescribed in their treatment protocol. Risk of developing neutropenia was lower when patients were treated with prophylactic colony-stimulating factor (CSF) and is also advised to be given alongside chemotherapy for regimens with an incidence of febrile neutropenia of 20% or higher [123]. Besides anthracyclines and platinum drug-containing regimens, treatment schedules containing alkylators and topoisomerase inhibitors can lead to myelosuppression [124].

Incidence of hematological toxicity in pediatric patients is scarcely reported. However, occurrence of this side-effect is reported for many patients with leukemia or lymphoma [125]. Due to the nature of these cancer types, e.g., infiltration of bone marrow by cancer cells, blood cell counts can be (very) low. As with chemotherapy-induced neutropenia, dose reduction might be required to prevent infection and further deterioration of the blood cell population.

### 2.6. Other Toxicities

#### 2.6.1. Ototoxicity

Chemotherapy associated with ototoxicity includes the platinum agents. Post-treatment hearing loss and/or tinnitus is mostly associated with cisplatin treatment and to a lesser extent with carboplatin or oxaliplatin [126]. Cisplatin-induced ototoxicity occurs in 23–50% of adults and more than 60% of children [127,128,129] and differs depending on genetic susceptibility [130]. Damage to the inner ear results mostly from ROS produced after cisplatin treatment [131]. Cisplatin is taken up via OCT2 and accumulates in hair cells, the stria vascularis, and spiral ganglion cells after which these structures are (irreversibly) destroyed leading to hearing loss mostly in the high frequency range [132]. Although aminoglycosides and furosemide do not belong to the group of drugs classified as chemotherapy, these medications do lead to ototoxicity and can aggravate the effects of platinum agents on the cochlea [133,134].

#### 2.6.2. Gastro-Intestinal Toxicity

Up to 80% of cancer patients receiving chemotherapy experience chemotherapy-induced gastrointestinal toxicity (CIGT), entailing pain sensation throughout the whole gastrointestinal tract, nausea, infection, and diarrhea [135]. CIGT likely arises as soon as the mucosal barrier covering the entire gastrointestinal tract is disrupted, leading to pro-inflammatory cytokine release, ulcerations, and inflammation, also known as mucositis [136]. This does not only cause cell death and dysfunction (malabsorption) of the gastrointestinal epithelium, but it also leads to a disbalance in the gut microbiome, which plays a large role in functioning of the immune system [135]. The pain, nausea, and dysphagia does not only severely decrease the quality of life of the patients, but the resulting malnutrition can also cause serious health issues or even death in fragile patients [137,138]. Chemotherapeutics that often generate severe CIGT and mucositis are methotrexate, 5-FU, cisplatin, and vinca alkaloids [137,139,140]. Toxicities such as mucositis, nausea, vomiting, dyspepsia, diarrhea, and abdominal pain occur in a large part of patients treated with radiotherapy [141,142].

#### 2.6.3. Gonadal Toxicity

Chemotherapy treatment can also lead to gonadal toxicity. In men, testicular damage can manifest in an acute but mild manner, mostly damaging developing spermatocytes. However, it can also cause a more severe toxicity, where also spermatogonial stem cells are damaged leading to an inability to repopulate the seminiferous tubules with sperm cells [143,144]. In women, the level of chemotherapy-induced ovarian toxicity is dependent on chemotherapy type, treatment protocol, and total cumulative dose [145]. There are several mechanisms via which chemotherapy can induce premature ovarian failure (POF) or insufficiency (POI). Direct ovarian toxicity, which is mostly caused by alkylating agents, is characterized by apoptosis of glomerular and stromal cells and direct damage to oocytes, and a decrease in primordial follicle pool by increased dormant follicle activation [145,146]. In addition, stromal and vascular damage is generated in the ovaries, mostly induced by anthracyclines, including fibrosis and vessel narrowing leading to decreased blood flow [145,146].

## 3. Conceptualization of Chemotherapeutics Driving Segmental Aging

As described above, toxicities to different organ systems are each inflicted via specific classes of chemotherapeutics. What is the reason for this and what determines the specificity for certain tissue types and subsequent severity of the side-effects? Drug-intrinsic properties such as biodistribution and reaction kinetics (partially) determine the target organs, whereas factors such as internalization and excretion of drugs by active pumps, (de)toxification mechanisms, and, e.g., the moment during the day the treatment is applied, influence outcome. Most anti-cancer therapies target the DNA at subcellular level, damaging the DNA by inducing different types of DNA lesions depending on the chemotherapy class (Figure 1). This does not only happen in cancer cells but also in the surrounding healthy tissue and more systemically throughout the body. Subsequently, DNA damage can result in (1) mutations and drive carcinogenesis or (2) impaired DNA replication, DNA and/or RNA synthesis, cell-cycle arrest, cellular senescence or cell death, and contribute to cellular or organ functional decline and accelerate aging. The different types of DNA damage can have different outcomes depending on, e.g., the affected organ or cell type (Table 1). Moreover, dosage will be an important determinant, as for example highlighted by animal experiments, showing acute lethality by high dose ionizing radiation. These mice initially die within a few days of gastrointestinal failure, as the stem cells in the crypts are the most rapidly dividing cells of our body, while with a lower dosage, bone marrow failure is observed after a couple of weeks [147,148,149]. As our organs comprise various cell types, differing in proliferation rate, differentiation status, and systemic hormonal and immunological parameters, one could imagine that this results in a variable susceptibility of cells to different types of DNA damage [150].

Under normal non-chemotherapy conditions, DNA damage already occurs continuously at a massive scale: namely about 10^5^ DNA lesions arise on a daily basis in every human cell [32,151,152]. Part of these lesions originate from exogenous sources, such as UV-light, oxidative damage, genotoxins from cigarette smoke, or even mechanical stress. However, most lesions likely arise from endogenous sources, such as reactive by-products of normal metabolism [151,153,154]. Together, this results in a broad variety of chemical modifications to our DNA, ranging from spontaneous deamination and hydrolysis or subtle base modifications to bulky helix distorting lesions, inter- and intrastrand DNA crosslinks, or single- and double-stranded DNA breaks [155,156]. Indirectly, also R-loop formation (hybridization of mature mRNA with double-stranded DNA), as a consequence of transcription stalling, can be considered as DNA damage. These and other aberrant DNA structures trigger a multifaceted network of DNA damage response signaling and lead to the subsequent removal of DNA lesions by various dedicated and complementary DNA repair processes [18,157]. These include amongst others homologous recombination (HR) and non-homologous end joining (NHEJ) for the repair of double-stranded DNA breaks [158], base excision repair (BER) for the removal of subtle base modifications [159], global genome-nucleotide excision repair (GG-NER) and transcription-coupled repair (TCR) for bulky helix distorting lesions [160], and interstrand crosslink (ICL) repair to resolve DNA crosslinks during replication [161] (Figure 2). Translesion synthesis polymerases can still bypass damaged DNA bases during replication giving rise to increased mutations over time [162].

If, however, the number of DNA lesions becomes too high for the repair machineries to tackle, patients treated with various DNA damaging treatments such as radio- and chemotherapy would be left with elevated levels of persisting DNA lesions. The chemotherapy side-effects observations made in Table 1 strengthen the hypothesis that elevated exposure would lead to accelerated aging later in life [12,15,156] and is in line with the DNA damage theory of aging [155,163]. Likewise, children with inborn mutations in DNA repair processes or nuclear integrity show segmental progeroid aging, affecting only a subset of tissues and organs [157,164,165,166]. Both the type of mutation and type of repair process affected in children with progeria syndromes and the classes of chemotherapeutics used for the treatment of (pediatric) cancers result in a specific accumulation of persisting DNA lesions and as such drive accelerated aging across various organs and tissues (Figure 3).

## 4. Conclusions

This review focused on the toxic side-effects of most classical chemotherapeutics (including radiotherapy) and the occurrence in different organs. It is also clearly displayed that most of the chemotherapeutics target and damage the DNA, resulting in impaired DNA replication, and DNA and RNA synthesis. These factors, together with the level of drug exposure and type of DNA or structural damage induced by the chemotherapeutic, all affect the degree of cancer therapy-induced toxicity. To improve quality of life of cancer patients and survivors, (further) reduction of treatment-induced side-effects is required. To achieve this, monitoring of patients is of great importance, especially during and between treatment cycles, but also regularly after, to timely observe development of toxicities. Health care personnel should be aware of the side-effects of the different components of cancer therapy received by each specific patient to be able to perhaps adjust therapy before (long-term) damage occurs. Additionally, screening patients for risk factors associated with certain therapies might prevent occurrence of certain toxicities, as the treatment protocol can possibly be adjusted beforehand, by, e.g., substituting for a less toxic drug. Lastly, identification of toxicity-related biomarkers could aid in better monitoring and early detection of overtreatment and side-effects, while these are potentially still reversible. Better insight into development of side-effects facilitates the implementation of nutritional or chemical interventions to reduce toxicity [9,156,167,168,169]. Patients can benefit from the aforementioned strategies, resulting in increased wellbeing, in both the short and long term.

## Figures and Tables

**Figure 1 cancers-14-00627-f001:**
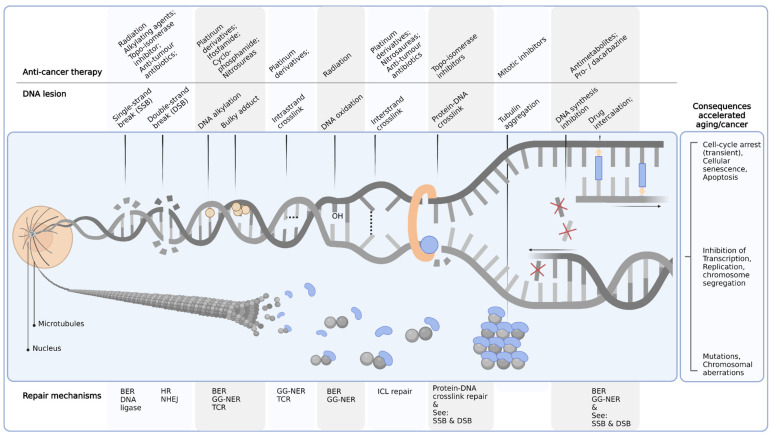
Chemotherapeutics, DNA damage, repair mechanisms, and consequences. Examples of commonly used anti-cancer treatments interfering with DNA metabolism (**top**); DNA lesions induced by these anti-cancer agents (**middle**); relevant DNA repair processes responsible for the removal of the lesions (bottom) and consequences of persisting DNA lesions accelerating aging and/or secondary cancers (**right**). BER, base excision repair; GG-NER, global genome-nucleotide excision repair; HR, homologous recombination; NHEJ, non-homologous end joining; TCR, transcription-coupled repair; ICL, interstrand crosslink; SSB, single-strand break; DSB, double-strand break.

**Figure 2 cancers-14-00627-f002:**
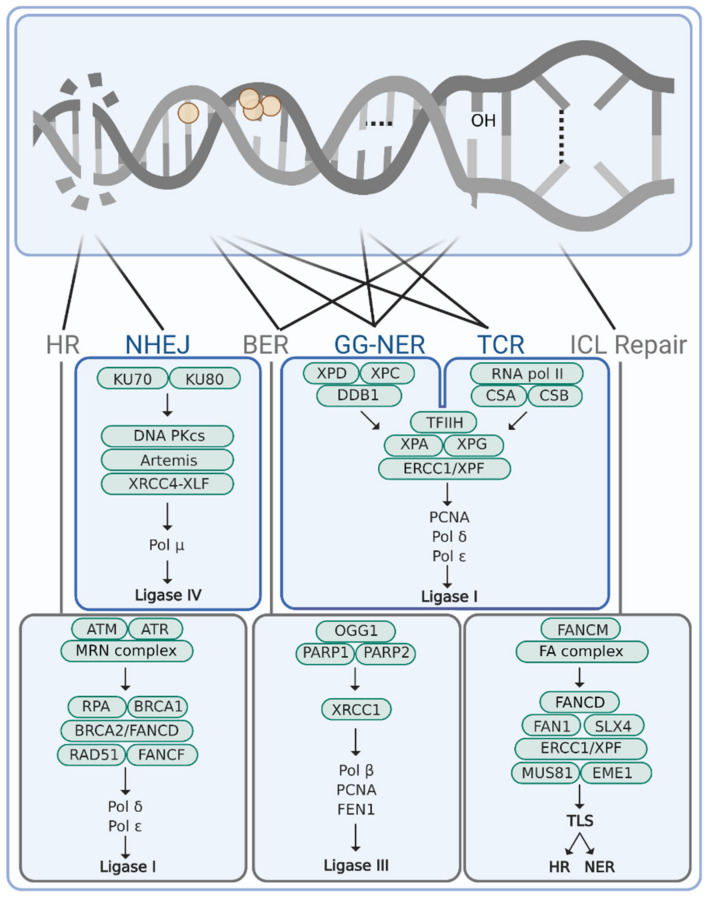
DNA damage repair mechanisms involved in repair of DNA lesions commonly caused by chemotherapeutics. Schematic representations of, from left to right, double-strand break, DNA alkylation, bulky adduct, intrastrand crosslink, DNA oxidation, and interstrand crosslink. DNA repair pathways responsible for the removal of the represented DNA lesions are schematically shown below with the key proteins involved in the depicted DNA repair processes. HR, homologous recombination; NHEJ, non-homologous end joining; BER, base excision repair; GG-NER, global genome-nucleotide excision repair; TCR, transcription-coupled repair; ICL, interstrand crosslink repair (during replication); TLS, translesion synthesis.

**Figure 3 cancers-14-00627-f003:**
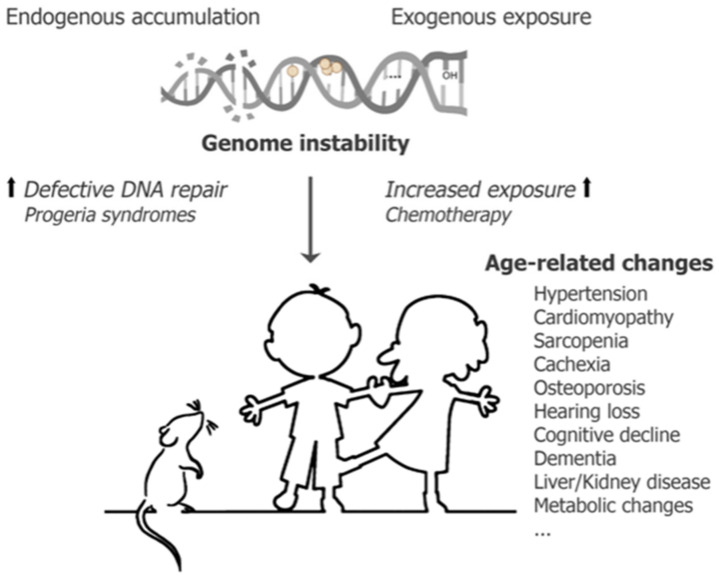
Genome instability induces premature aging. Persisting DNA damage, originating from both endogenous and exogenous sources, normally results in the onset of aging pathologies. These consequences are accelerated either when DNA repair processes are defective, in the case of progeria syndromes and thus more endogenous lesions persist, or when exogenous exposure is increased as during chemotherapy treatment.

**Table 1 cancers-14-00627-t001:** Side-effects of anti-cancer therapy observed directly or late in life.

Toxicity	Chemotherapy		Toxicity Details
**Nephrotoxicity**	**Alkylating agents**	Platinum derivatives	
		Cisplatin [20,21,22,23,24,25,26,27,28,29,30]	FS [21,25,26,27,28,29], salt wasting [21,24,26,27,28,29,30], hypomagnesemia [21,23,24,25,26,27,28,29,30,31], NDI [22,26,28,29,32], AKI [21,24,25,26,27,28,29,30,31], CKD [21,26,28,29], distal renal tubular acidosis [21,24,25,26,28], TMA [21,25,26,28], erythropoietin deficiency [21,26], renal concentrating defect [25], transient proteinuria [21], glomerular disease [30], tubulointerstitial disease [30], HUS [30], chronic renal failure [21]
		Carboplatin [24,30,31]	Hypomagnesemia [24], Glomerular disease [30], Tubulointerstitial disease [30]
		Ifosfamide [21,24,27,29,30,31]	FS [21,24,27,29,30], NDI [24,27,29], salt wasting [21], hyponatremia-SIADH [24], hemorrhagic cystitis [24,27,31], AKI (ATN) [21,27,29], CKD [27], renal tubular acidosis [24], glomerular disease [21,30], tubulointerstitial disease [30]
		Cyclophosphamide [24,29,31]	Hemorrhagic cystitis [24,31], NDI [29]/hyponatremia-SIADH [24]
		Melphalan [24,30]	Hyponatremia-SIADH [24], glomerular dysfunction [30]
	**Antimetabolites**	Methotrexate [21,24,27,29,30,31,33,34]	Crystal nephropathy [24,27,29,31,33,34], AKI [21,24,27,30], renal tubular obstruction [30,33], CKD [29], acute tubular necrosis [21], hyponatremia-SIADH [24], hypomagnesemia [30], hypokalemia [30], hypocalcemia [30], glomerular disease [30,33], tubulointerstitial disease [30]
		Gemcitabine [21,24,27,29,30,33,35]	TMA [21,27,29,33,35], HUS [27,29,30,33,35], AKI [24,27,29]
		Pemetrexed [24,27,30,33]	AKI (ATN) [24,27,33], AIN [27], renal tubular acidosis [24,27,33], NDI [24,27,33], glomerular disease [30]
		Nitrosoureas [24,29,30,31]	CKD [24,29] (chronic interstitial nephritis, glomerulosclerosis)
		Streptozocin [30]	FS [30]
		Anthracyclines	
		Doxorubicin [33,36]	Focal segmental glomerular sclerosis [33], TMA [33], AKI [33]
	**Anti-cancer antibiotics**	Mitomycin [30,33,35]	HUS [30,33,35], prerenal azotemia [30], glomerular disease [30], TMA [35]
		Plicamycin [30]	Glomerular disease [30]
**Hepatotoxicity**	**Alkylating agents**	Platinum derivatives	
		Oxaliplatin [37,38,39,40,41]	Fatty liver/steatosis [38,39,40], VOD [38,39,40], SOS [37,40,41], nodular hyperplasia [38], fibrosis [37,38]
		Cisplatin [28,39,42,43]	Sinusoidal dilation and obstruction [28,42], necrosis [42], infiltration of inflammatory cells [42], steatosis [39], cholestasis [39]
		Carboplatin [39]	Cholestasis [39], VOD [39]
		Cyclophosphamide [38,39,43]	VOD [38], hepatitis [39]
		Melphalan [38,39,43]	Hepatitis [38,39], VOD [38]
		Busulfan [31,38,39,43]	Hepatitis [38], cholestasis [38,39], VOD [38,40]
		Nitrosoureas [38,40,43]	Hepatitis [38]
		Carmustine [39]	Hepatitis [39]
		Lomustine [39]	Hepatitis [39]
		Chlorambucil [38,39,43]	Cholestasis [38,39]
		Dacarbazine [39]	VOD [39]
		Procarbazine [39]	Hepatitis [39]
	**Topoisomerase inhibitors**	Irinotecan [37,38,39,40,43]	Fatty liver/steatosis [37,38,39,40,43], hepatitis [37,43], VOD [39]
		Etoposide [39,44]	Hepatitis [39], cholestasis [39], hepatocellular injury [44], hyperbilirubinemia [44]
	**Enzyme**	Asparaginase [38,39]	Steatosis [38,39], hepatitis [38,39]
	**Antimetabolites**	Cytarabine [38,39,40,43]	Biliary stricture [38,39,43], cholestasis [38,39,43,44], fibrosis [43]
		5-Fluorouracil [37,38,39,40,41]	Steatosis [37,38], hepatitis [38,39], VOD [38], hyperbilirubinemia [37], cirrhosis [41]
		Methotrexate [38,39,40,43,44]	Hepatitis [38,39], cirrhosis [38,39,43,44], nodular hyperplasia [38], fibrosis [38,39,43,44]
		Anthracyclines [38,40]	Hepatitis [38], cholestasis [38], VOD [38,40]
		Doxorubicin [39]	Hepatitis [39], cholestasis [39], VOD [39]
		Capecitabine [37,39]	Cholestasis [39], steatosis [37]
		Floxuridine [39,41]	Hepatitis [39], cholestasis [39], bile duct sclerosis [41]
		Gemcitabine [39]	Hepatitis [39], cholestasis [39]
		Mercaptopurine [38,39,43,44]	VOD [39,43], cholestasis [38,43,44], jaundice [38], hepatocellular injury [43,44], hepatitis [38,44]
	**Anti-cancer antibiotics**	Dactinomycin [39]	Hepatitis [39], VOD [39]
		Mitomycin [39]	Hepatitis [39], VOD [39]
		Mitoxantrone [39]	Hepatitis [39]
	**Mitotic inhibitors**	Vinca alkaloids	
		Vincristine [39]	Hepatitis [39]
		Vinblastine [39,43]	Hepatitis [43], VOD [39]
		Radiotherapy [43,44,45]	RILD [45], VOD [43,44,45], combined modality-induced liver damage (CMILD) [45], SOS [45]
**Neurotoxicity**	**Alkylating agents**	Platinum derivatives [46,47]	Distal symmetrical sensory impairment [46,47], ataxia [46,47], (peripheral) neurotoxicity [46,47], loss of deep tendon reflexes [47]
		Cisplatin [48,49,50]	Acute encephalopathy [49], chronic leukoencephalopathy (if combined with radiotherapy) [49], posterior reversible (leuko)encephalopathy [48,49], stroke/arterial ischemia [48,49], seizures [48,49], SIADH [48,49], transient cortical blindness [48,49], myelopathy [48,49], peripheral neuropathy [48], sensory impairment [48,50], sensory ataxia [48], myasthenic syndrome [48], muscle cramps [48], Lhermitte symptom [48]
		Carboplatin [48,49]	Posterior reversible (leuko)encephalopathy [49], peripheral neuropathy [47,48], acute (cold-induced) transient syndrome [47], paresthesiasis in distal extremities and perioral region [47], chronic cumulative sensory neuropathy [47]
		Oxaliplatin [46,47,48,49,50]	Posterior reversible (leuko)encephalopathy [49], acute (peripheral) neurotoxicity [46,48,50], chronic sensory neuropathy [48,50], acute cramps and fasciculations [50]
		Cyclophosphamide [49]	Posterior reversible (leuko)encephalopathy [49], stroke/arterial ischemia [49], SIADH [49]
		Nitrosoureas	
		Carmustine [49]	Acute encephalopathy [49]
		Busulfan [49]	Seizures [49]
		Chlorambucil [49]	Acute encephalopathy [49], seizures [49]
		Procarbazine [48,49]	Acute encephalopathy [48,49], paresthesiasis [48], myalgia [48]
		Ifosfamide [49]	Acute encephalopathy [49], posterior reversible (leuko)encephalopathy [49], seizures [49], extrapyramidal syndrome [49]
		Thiotepa [49]	Acute encephalopathy [49], myelopathy [49]
	**Topoisomerase inhibitors**	Etoposide [49]	Acute encephalopathy [49], chronic leukoencephalopathy [49], seizures [49], transient cortical blindness [49]
	**Antimetabolite**	Methotrexate [48,49]	Acute encephalopathy [48,49], chronic leukoencephalopathy [48,49], posterior reversible (leuko)encephalopathy [49], stroke/arterial ischemia [48,49], focal neurologic deficits [49], seizures [49], myelopathy [49]
		Cytarabine [48,49]	Acute encephalopathy [48,49], posterior reversible (leuko)encephalopathy [49], acute pancerebellar syndrome * [48,49], extrapyramidal syndrome [48,49], peripheral neuropathies [48], brachial plexopathy [48], myelopathy (liposomal cytarabine) [49], Horner syndrome [48], lateral bulbar palsy [48], nystagmus [48], lethargy [48], aseptic meningitis [48], locked-in syndrome [48]
		Gemcitabine [49]	Acute encephalopathy [49], posterior reversible (leuko)encephalopathy [49], thrombotic microangiopathy [49], seizures [49]
		5-Fluorouracil [49]	Acute encephalopathy [49], acute pancerebellar syndrome [49], stroke/arterial ischemia [49], extrapyramidal syndrome [49]
		Capecitabine [49]	Acute encephalopathy [49], acute pancerebellar syndrome [49]
		Fludarabine [48,49]	Acute encephalopathy [49], chronic leukoencephalopathy [48,49], transient cortical blindness [48,49], seizures [48], paresthesiasis [48], ataxia [48], paralysis [48]
		Pentostatin [49]	Acute encephalopathy [49], chronic leukoencephalopathy [49]
		Nelarabine [48,49]	Acute encephalopathy [49], seizures [48,49], distal sensory impairment [48], distal motor impairment [48], myelopathy [49], tremor [48], muscle weakness [48]
		Hydroxyurea [49]	Chronic leukoencephalopathy [49], seizures [49]
		Anthracyclines	
		Doxorubicin [49]	Extrapyramidal syndrome [49]
	**Anti-cancer antibiotics**	Mitomycin [49]	Acute encephalopathy [49]
		Bleomycin [48]	Cerebral and myocardial infarcts [48]
	**Enzyme**	Asparaginase [49]	Acute encephalopathy [49], intracranial hemorrhage [49], sinus/cortical vein thrombosis [49]
	**Mitotic inhibitors**	Vinca alkaloids	
		Vincristine [46,48,49,50] (and Vinblastine, Vindesine, Vinorelbine) [48]	Acute encephalopathy [48,49], posterior reversible (leuko)encephalopathy [49], acute pancerebellar syndrome [49], seizures [48,49], SIADH [48,49], transient cortical blindness [48,49], sensory impairment [46,48,50], distal motor impairment [46,48], ataxia [46,48], autonomic neuropathy [46,48,50], muscle cramps [48,50], mild distal weakness [48,50], parkinsonism [48], athetosis [48]
		Vinflunine [49]	Posterior reversible (leuko)encephalopathy [49]
		Taxanes	
		Paclitaxel [46,47,49,50]	Distal sensory impairment [46,50], AMS [46,50], myopathy [50], distal motor impairment [46]
		Docetaxel [47,50]	Sensory impairment [50], myalgia [50], myopathy [50]
		Epothilones [46]	Spinal cord injury [46], distal sensory impairment [46], distal motor impairment [46]
		Eribulin [46]	Sensory impairment [46], motor impairment [46]
	**Proteasome inhibitors**	Bortezomib [46,48,50]	Severe neuropathic pain [46,48,50], autonomic neuropathy [50], dizziness [48], aphasia [48]
		Radiotherapy [49] (combined with chemotherapy)	Chronic leukoencephalopathy [49], myelopathy [49]
**Cardiotoxicity**	**Alkylating agents**	Cisplatin [42,51,52,53,54,55]	Degeneration and necrosis of cardiac muscle [42], fibrosis [42], vacuolated cytoplasm [42], blood infiltration [42], palpitations [51,52], left-sided chest pain [51,52]/angina pectoris [54], hypotension [51,54], arrhythmias [51,53,54], interventricular block [51], MI [51,52,53,54], atrial fibrillation [53], left bundle branch block [53], ischemia [52], hypertension [52,54], thromboembolism [54,55], myocarditis [54], CHF [54], cardiomyopathy [54]
		Mitomycin [51,52,54]	CHF [51,54], HF [52], cardiomyopathy [52],
		Carmustine [51]	Chest pain [51], hypotension [51], arrhythmia [51]
		Busulfan [51]	CHF [51], palpitations [51], cardiac tamponade [51], pulmonary congestion [51], cardiomegaly [51], pericardial effusion [51]
		Chlormethine [51]	Persistent tachycardia [51], pulse irregularity [51], junctional or atrial ectopic beats [51]
		Amsacrine [53]	Arrhythmia [53], CHF [53], cardiomyopathy [53]
		Capecitabine [55]	Ischemia [55]
		Cyclophosphamide [51,53,54,55,56,57]	Neurohumoral activation [56,57], mitral regurgitation [56,57] CHF [51,53], chest pain [51], pleural and pericardial effusions [51], pericardial friction rub [51], cardiomegaly [51] myo(peri)carditis [53,54], HF [52,54,57], cardiomyopathy [52], left ventricular dysfunction [54,55], arrhythmia [54]
		Ifosfamide [51,54,55]	CHF [51,54], pleural effusion [51], arrhythmia [51,54], left ventricular dysfunction [55]
	**Antimetabolites**	Anthracyclines [51,52,53,54,56,57]	CHF [53,56], left ventricular dysfunction [53,56,57], (acute) myocarditis [56,57], arrhythmia [53,56,57]
		Doxorubicin [52,53,57,58]	Reversible acute myopericarditis [53], HF [52,57], CHF [58], left ventricular dysfunction [55]
		Doxorubicin, Daunorubicin [51]	Reversible acute myopericarditis [51], HF [51]
		Epirubicin [51,58]	Arrhythmia [51], pericarditis [51], myocarditis [51], CHF [51,58], left ventricular dysfunction [55]
		Idarubicin [51,55]	CHF [51], left ventricular dysfunction [55], arrhythmias [51], angina pectoris [51], MI [51]
		Mitoxantrone [51,54]	Arrhythmias [51,54], CHF [51,54], MI [51,54], HF [52]
		Capecitabine, 5-fluorouracil, cytarabine [56]	Ischemia [56], pericarditis [56], CHF [56], cardiogenic shock [56]
		Capecitabine, 5-fluorouracil [54]	Angina-like chest pain [54], MI [54], arrhythmia [54], HF [54]
		5-Fluorouracil [51,52,53,55,57]	Angina pectoris [51,52,53], MI [51,52,55], hypotension [51,53], cardiogenic shock [51,53,55,57], left ventricular dysfunction [53], HF (with global hypokinesis [53]) [55,57], ischemia [52,55,57], pericarditis [57], arrhythmia [55]
		Cytarabine [53]	Arrhythmia [53], pericarditis [53], acute respiratory distress [53], CHF [53]
		Clofarabine [55]	Left ventricular dysfunction [55]
	**Topoisomerase inhibitors**	Etoposide [51,52]	Hypotension [51,52], MI [51]
		Teniposide [51]	Arrhythmia [51], hypotension [51]
	**Mitotic inhibitors**	Paclitaxel, vinca alkaloids [56,57]	Sinus bradycardia [56,57], ventricular tachycardia [56,57], atrioventricular block [56,57], hypotension [56], CHF [56], ischemia [56,57], HF [57]
		Paclitaxel [51,52,53,54,55]	Arrhythmia [51,52,53,54,55], MI [51,53,55], atrioventricular or left bundle branch block [51,54], second- or third-degree heart block [53], cardiac ischemia [53,54,55], heart block [52]
		Vinca alkaloids [51,52]	MI [51,52], dyspnea [51], pulmonary oedema [51], atrial fibrillation [51], angina pectoris [52]
		Docetaxel [54,55]	Left ventricular dysfunction [55], ischemia [54,55], MI [55]
	**Other**	Tretinoin [51]	Retinoic acid syndrome ** [51], arrhythmia [51], hypotension [51], hypertension [51], HF [51]
		Pentostatin [51]	Angina pectoris [51], MI [51], CHF [51], arrhythmia [51]
**Hematological Toxicity**	**Alkylating agents**	Platinum derivatives	
		Carboplatin [59,60]	Thrombocytopenia [59,60]
		Oxaliplatin [60]	Thrombocytopenia [60]
		Nitrogen mustard [61,62]	Bone marrow suppression [61], myelosuppression [62]
		Melphalan [61]	Bone marrow suppression [61]
		Cyclophosphamide [34,61]	Myelosuppression (leukopenia)/neutropenia [34], anemia [34], bone marrow suppression [34,61]
		Chlorambucil [61,63]	Bone marrow suppression [61,63]
		Nitrosoureas [61,62]	Bone marrow suppression [61], myelosuppression [62]
		Dacarbazine [62]	Myelosuppression [62]
		Busulfan [62]	Myelosuppression [62]
	**Antimetabolites**	Methotrexate [61]	Bone marrow suppression [61]
		Antipyrimidines [62]	Myelosuppression [62]
		5-Fluorouracil [61]	Bone marrow suppression [61]
		Cytarabine [61]	Bone marrow suppression [61]
		6-Mercaptopurine [61]	Bone marrow suppression [61]
		Anthracyclines [62]	Myelosuppression [62]
		Doxorubicin [61]	Bone marrow suppression [61]
	**Anti-cancer antibiotics**	Dactinomycin [61]	Bone marrow suppression [61]
	**Combination therapies**	Docetaxel-cyclophosphamide [64]	Febrile neutropenia [64]
		Cyclophosphamide-doxorubicin—vincristine-prednisone [65]	Neutropenia [65], febrile neutropenia [66]
		Cyclophosphamide—methotrexate—5-fluorouracil [67]	Leukopenia [67], anemia [67], neutropenia [67], thrombocytopenia [67]
		Doxorubicin-cyclophosphamide [65]	Neutropenia [65]
		Cyclophosphamide-doxorubicin-5-fluorouracil [65]	Neutropenia [65]
		5-Fluorouracil-cisplatin [65]	Neutropenia [65]
		Etoposide-doxorubicin-5-fluorouracil [65]	Neutropenia [65]
		Docetaxel-doxorubicin [65]	Neutropenia [65]

FS = Fanconi syndrome; NDI = nephrogenic diabetes insipidus; AKI = acute kidney injury; CKD = chronic kidney disease; TMA = thrombotic microangiopathy; HUS = hemolytic uremic syndrome; SIADH = syndrome of inappropriate anti-diuretic hormone secretion; ATN = acute tubular necrosis; AIN = acute interstitial nephritis; VOD = veno-occlusive disease; SOS = sinusoidal obstruction syndrome; RILD = radiation-induced liver damage; AMS = arthralgia-myalgia syndrome; CHF = congestive heart failure; MI = myocardial infarction; HF = heart failure; * acute pancerebellar syndrome is characterized by ataxia, dysarthria, and oculomotor alterations [68]; ** retinoic acid syndrome is characterized by fever, respiratory distress, bodyweight gain, peripheral oedema, pleural-pericardial effusions, and MI [51].

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
