# Peer review of "Chemotherapy Side-Effects: Not All DNA Damage Is Equal"

_cancers, 2022, doi:10.3390/cancers14030627_

Round 1
Reviewer 1 Report
van den Boogaard et al in the manuscript of ‘Chemotherapy side‐effects: not all DNA damage is equal’ (Cancers-1563221) outlined the (mostly chronic or long-term) side-effects of chemotherapies as well as radiotherapy. The review article specifically summarized the main organs/tissues targeted by the chemo- or radio-therapies, and briefed the cellular, molecular and sub-cellular mechanisms of the side-effects or undesired toxicities in the cancer patients treated.
The authors reviewed a good volume of literature, tabled the anticancer therapeutics and organ/tissue-specific side effects shoulder by shoulder, and illustrated the mechanisms of the side effects.
Overall, the manuscript provided a comprehensive profile of updated information regarding chemo-/radio-therapy side effects useful for evaluating future clinical use of preventive and/or interventive measures to minimize the side effects and optimize patient life quality.
Minor concerns are listed below-
- Line 14- “…but also harm neighboring healthy cells..”, the emphasis on “neighboring” is not correct or necessary, in that chemodrugs are almost always administered by i.v. and circulated in the blood throughout the body.
- Line 51-“a.o. tumor type” is hard to understand.
- Line 54-“secondary malignancies”. Is the “secondary’ right word here? Perhaps, “therapeutic-induced” is more accurate.
Author Response
We want to thank the reviewers for their time and helpful suggestions and have accordingly made several adjustments that raised the quality of our manuscript. Please find below our point-by-point answers indicating the changes we have made.
Reviewer 1:
van den Boogaard et al in the manuscript of ‘Chemotherapy side‐effects: not all DNA damage is equal’ (Cancers-1563221) outlined the (mostly chronic or long-term) side-effects of chemotherapies as well as radiotherapy. The review article specifically summarized the main organs/tissues targeted by the chemo- or radio-therapies, and briefed the cellular, molecular and sub-cellular mechanisms of the side-effects or undesired toxicities in the cancer patients treated.
The authors reviewed a good volume of literature, tabled the anticancer therapeutics and organ/tissue-specific side effects shoulder by shoulder, and illustrated the mechanisms of the side effects.
Overall, the manuscript provided a comprehensive profile of updated information regarding chemo-/radio-therapy side effects useful for evaluating future clinical use of preventive and/or interventive measures to minimize the side effects and optimize patient life quality.
Minor concerns are listed below
Line 14- “…but also harm neighboring healthy cells..”, the emphasis on “neighboring” is not correct or necessary, in that chemodrugs are almost always administered by i.v. and circulated in the blood throughout the body.
We thank the reviewer for critically reading our manuscript and agree that for chemotherapeutics delivered via i.v. the word neighboring is indeed unnecessary. As such we have removed this word from the simple summary.
Line 51-“a.o. tumor type” is hard to understand.
Thank you for your question. Different tumor types usually require different treatment protocols differing in chemotherapeutics used and timing of administration. To further clarify the wording, we have adjusted this sentence to: “More chemotherapeutic drugs were developed during the 20th century, which are nowadays used in different combinations in various treatment schedules depending on a.o. tumor type, risk stratification group and genetic make‐up” (line 50-53).
Line 54-“secondary malignancies”. Is the “secondary’ right word here? Perhaps, “therapeutic-induced” is more accurate.
We agree with the reviewer and indeed mean with secondary malignancies cancers caused by the treatment with radiation or chemotherapy. As the term secondary malignancies is commonly used, we adjusted wording to “secondary (therapeutic-induced) malignancies” (line 56/57).
Reviewer 2 Report
Authors have done an impressive amount of literature search and wrote the review with a good amount of information. I would like to suggest authors should include one figure and explanation of the figure related to double strand break repair and base exchange repair followed by proteins such BRACA, ATM/ATR, XPC, XPA, DDB1 etc as a mechanism in the figure. I won’t influence the direction of the figure, but I believe adding mechanisms with key candidates will significantly improve the paper.
Author Response
We want to thank the reviewers for their time and helpful suggestions and have accordingly made several adjustments that raised the quality of our manuscript. Please find below our point-by-point answers indicating the changes we have made.
Reviewer 2:
Authors have done an impressive amount of literature search and wrote the review with a good amount of information. I would like to suggest authors should include one figure and explanation of the figure related to double strand break repair and base exchange repair followed by proteins such BRACA, ATM/ATR, XPC, XPA, DDB1 etc as a mechanism in the figure. I won’t influence the direction of the figure, but I believe adding mechanisms with key candidates will significantly improve the paper.
Thank you for your valuable question. We indeed agree that by adding a figure mechanistically explaining some of the key DNA repair processes, the context of that subsequent effects of accumulation of persisting distinct DNA lesions becomes clearer to a broader audience as multiple readers would perhaps not exactly know how DNA lesions are removed by these specific DNA repair processes. To make space for this figure we have now moved Figure 1 to the end of the introduction (closest to where first cited) and incorporated a new Figure 2 at its previous location, now on page 18. The additional description and citation to this figure and furthermore dedicated reviews on the specific DNA repair processes is included on page 19 lines 523-531: “These and other aberrant DNA structures trigger a multifaceted network of DNA damage response signaling and lead to the subsequent removal of DNA lesions by various dedicated and complementary DNA repair processes [18, 158]. These include amongst others homologous recombination (HR) and non-homologous end joining (NHEJ) for the repair of double stranded DNA breaks [159], base excision repair (BER) for the removal of subtle base modifications [160], global genome-nucleotide excision repair (GG-NER) and transcription coupled repair (TCR) for bulky helix distorting lesions [161] and interstrand crosslink (ICL) repair to resolve DNA crosslinks during replication [162] (Figure 2).”